# The Role of Cholangioscopy in Biliary Diseases

**DOI:** 10.3390/diagnostics13182933

**Published:** 2023-09-13

**Authors:** Aurelio Mauro, Stefano Mazza, Davide Scalvini, Francesca Lusetti, Marco Bardone, Pietro Quaretti, Lorenzo Cobianchi, Andrea Anderloni

**Affiliations:** 1Gastroenterology and Endoscopy Unit, Fondazione IRCCS Policlinico San Matteo, 27100 Pavia, Italya.anderloni@smatteo.pv.it (A.A.); 2Department of Internal Medicine, University of Pavia, 27100 Pavia, Italy; 3Unit of Interventional Radiology, Department of Radiology, Fondazione IRCCS Policlinico San Matteo, 27100 Pavia, Italy; 4Department of General Surgery, Fondazione IRCCS Policlinico San Matteo, 27100 Pavia, Italy

**Keywords:** cholangioscopy, direct peroral cholangioscopy, indeterminate biliary stricture, difficult bile stones, intrahepatic stones, Mirizzi syndrome, artificial intelligence, surgical cholangioscopy, percutaneous cholangioscopy, hepaticogastrostomy

## Abstract

Endoscopy plays a central role in diagnostic and therapeutic approaches to biliary disease in both benign and malignant conditions. A cholangioscope is an endoscopic instrument that allows for the direct exploration of the biliary tree. Over the years, technology has improved endoscopic image quality and allowed for the development of an operative procedure that can be performed during cholangioscopy. Different types of instruments are available in this context, and they can be used in different anatomical access points according to the most appropriate clinical indication. The direct visualization of biliary mucosa is essential in the presence of biliary strictures of unknown significance, allowing for the appropriate allocation of patients to surgery or conservative treatments. Cholangioscopy has demonstrated excellent performance in discriminating malignant conditions (such as colangiocarcinoma) from benign inflammatory strictures, and more recent advances (e.g., artificial intelligence and confocal laser endomicroscopy) could further increase its diagnostic accuracy. Cholangioscopy also plays a primary role in the treatment of benign conditions such as difficult bile stones (DBSs). In this case, it may not be possible to achieve complete biliary drainage using standard ERCP. Therapeutic cholangioscopy-guided lithotripsy allows for stone fragmentation and complete biliary drainage. Indeed, other complex clinical situations, such as patients with intra-hepatic lithiasis and patients with an altered anatomy, could benefit from the therapeutic role of cholangioscopy. The aim of the present review is to explore the most recent diagnostic and therapeutic advances in the roles of cholangioscopy in the management of biliary diseases.

## 1. Introduction

Digestive endoscopy was developed as an evolution of fluoroscopic gastrointestinal examinations (e.g., barium enema and barium swallow), allowing for the direct visualization of the gastrointestinal tract with the possibility of performing diagnostic and operative procedures. Biliopancreatic endoscopy performed via endoscopic retrograde cholangiopancreatography (ERCP) is an operative procedure that functions with the help of the diagnostic features of cholangiopancreatography. However, as in fluoroscopic gastrointestinal examinations, the pathological findings of cholangiopancreatographies (e.g., biliary strictures) are only moderately visible. Moreover, fluoroscopically guided biliary brushing and/or biopsies have sub-optimal diagnostic yields [1]. Thus, there is a need to endoluminally explore the biliopancreatic tract with dedicated endoscopic devices in challenging situations.

Peroral cholangioscopy (POC) is a technique that allows for the direct visualization of the biliary tree; it was introduced early in the 1970s with the so-called “mother-baby” system consisting of a videocholangioscope, i.e., the baby scope, which is inserted into the accessory channel of the “mother” duodenoscope. This type of system requires two operators, thus limiting its maneuverability. The currently available videocholangioscope (Olympus Medical Systems, Tokyo, Japan) has been shown to improve the quality of images and is also applicable to Narrow-Band Imaging (NBI) systems [2].

In 2007, a new single-operator cholangioscope (SOC) was introduced with a disposable fiberoptic scope (SpyGlass, Boston Scientific Endoscopy, Marlboro, MA, USA). The first version of the SOC produced sub-optimal images, but in 2015, a digital version of the SpyGlass SOC (DSOC) was developed, leading to enhanced image quality and a wider field of view [3,4]. The SOC consists of a disposable delivery catheter of 10 Fr in diameter that is capable of four-way deflected steering and has an outer diameter of 3.3 mm, an accessory channel of 1.2 mm, and separate, dedicated irrigation channels (Figure 1).

As in the “mother-baby” system, the digital SOC is inserted through the accessory channel of a conventional duodenoscope, but it is attached close to the working channel of the duodenoscope, thereby improving the maneuverability and the comfort of the operator. More recently, another digital cholangioscope providing a full high-definition view (eyeMAX™, Micro-Tech™, Nanjing, China) was made available in routine practice [5].

Another technique involves the use of the peroral direct cholangioscope (D-POC), which was first described in 2006 [6]. It employs an ultra-slim gastroscope, which was originally developed for use in pediatric patients and for transnasal applications. The advantages of the D-POC are a wider availability of the device in endoscopic services, the ability to obtain high-quality images, and the possibility of using image-enhanced function systems. On the other hand, it is more difficult to insert the scope if the common bile duct (CBD) is not sufficiently dilated, and it is sometimes difficult to stabilize its position; however, the insertion of the D-POC with a balloon catheter on a guidewire introduced in the common bile duct can aid the insertion of the scope [2].

Cholangioscopy is usually performed using the per-oral route, but the small caliber of the device also allows for its application through the percutaneous and intra-operative routes via trans-hepatic and trans-cystic insertion, respectively. Moreover, a novel short DSOC (65 cm in length) (SpyGlass™ Discover, Boston Scientific Endoscopy, Marlboro, MA, USA) was recently developed that permits ergonomic navigation throughout the biliary system. The main characteristics of the currently available cholangioscopy systems are summarized in Table 1.

All these cholangioscopes have the possibility to have different accessories inserted inside the working channel (e.g., biopsy forceps, retrieval baskets and snares, dilation balloon catheters), allowing operative procedures. Indeed, POC has gained popularity for the treatment of complex choledocholithiasis (e.g., large stones and intrahepatic stones) with the use of cholangioscopy-assisted lithotripsy, which aids in complete bile duct clearance. Direct exploration of the biliary tree also has other applications, ranging from retrieval of migrated stents and exploration of the cystic duct in the case of Mirizzi syndrome to the evaluation of the biliary mucosa after radiofrequency.

In this review, we summarize the diagnostic and operative applications of cholangioscopy, according to the latest literature evidence and current clinical practice guidelines.

## 2. Indeterminate Biliary Strictures

One of the main applications of POC is the evaluation of biliary strictures (BSs). Almost 20% of BSs are of indeterminate etiology at their presentation [7]. Indetermined BSs are defined when cross-sectional imaging, as well as tissue sampling, is inconclusive or negative, and this represents a challenging clinical scenario [8]. BSs located at the biliary hilum require a multidisciplinary approach for diagnostic and therapeutic decisions [9]. Indeed, almost two-thirds of indeterminate BSs are malignant but one-quarter of BSs at the time of surgical resection are benign [10,11] (see Table 2). Therefore, preoperative evaluation is crucial to properly refer patients for oncological/surgical treatment or conservative treatment in the case of malignant or benign disease, respectively.

Magnetic resonance cholangiopancreatography (MRCP) is generally the cross-sectional technique of choice for the evaluation of BSs, especially for those located in the hepatic hilum or those that are intrahepatic, as it has shown superiority over CT in this setting [12]. However, a meta-analysis by Romagnuolo et al. of 4711 patients showed a sub-optimal sensitivity of 88% for detecting malignancies [13]. Non-invasive biomarkers, such as carbohydrate antigen 19-9 (CA 19-9), are also not helpful to differentiate benign and malignant BSs [14].

### 2.1. Endoscopic Sampling with ERCP and EUS-FNB

Histological diagnosis with endoscopic sampling is needed when non-invasive tests are inconclusive about the nature of BSs. Endoscopic sampling during ERCP is usually the first-line approach for proximal strictures, whereas endoscopic ultrasound fine needle biopsies (EUS-FNB) are performed as the first approach in the case of distal strictures [15]. However, both procedures have some limitations and sub-optimal diagnostic yields. ERCP-guided brush cytology and forceps biopsies are historically limited by a low sensitivity (45% and 48.1%, respectively) and the combination of the two techniques increases the sensitivity only modestly (59.4%) [1], even after optimization of the technique by increasing the number of brushing passes and biopsies [16,17] and improving device design [18].

EUS-guided tissue acquisition is increasingly used for the diagnosis of biliary strictures caused by obstructive tumor masses (e.g., pancreatic cancer and mass-forming cholangiocarcinoma). When FNB needles became available on the market, this allowed excellent sensitivity and specificity to detect malignancy in distal lesions [19]. However, in the case of strictures secondary to endoductal vegetation, strictures located in the biliary hilum, or in the presence of a biliary stent, the performance of EUS-FNB is impaired [20].

### 2.2. Per-Oral Cholangioscopy

Recent guidelines suggest the use of POC when tissue acquisition with the standard biliopancreatic endoscopy techniques fails to reach a definitive diagnosis, especially in the case of a proximal stricture [15]. POC has great advantages over other endoscopic techniques in allowing the direct visualization of the BS, evaluating the characteristics of the surrounding mucosa, and performing targeted biopsies. Diagnostic POC is a safe procedure that does not increase the expected adverse events that are encountered during ERCP (e.g., pancreatitis, cholangitis, bleeding, and perforation); a meta-analysis showed a pooled rate of POC-related adverse events of 7% [21], which is similar to the expected rate of post-ERCP complications [22].

The mucosa of malignant BSs is characterized by peculiar aspects such as abnormal vessels, nodulations and vegetations, friability, and an irregular surface with or without ulcerations. Different classifications of the mucosal pattern of BSs have been proposed in recent years in order to differentiate benign and malignant BSs [23,24]. Recently, Kahaleh et al. proposed a novel classification: the Mendoza classification [25]. In their study, 14 experienced endoscopists reviewed images and clips of DSOC using five criteria (the presence of tortuous and dilated vessels, the presence of irregular nodulations, the presence of raised intraductal lesions, the presence of an irregular surface with or without ulcerations, and the presence of friability) derived from a previous expert consensus. The intraclass correlation coefficient was very high for tortuous and dilated vessels (0.86), raised intraductal lesions (0.90), and presence of friability (0.83), while it was moderate for the presence of an irregular surface with or without ulcerations (0.44). Moreover, the diagnostic intraclass correlation was almost perfect for neoplastic (0.90) and non-neoplastic (0.90) diagnoses. The overall diagnostic accuracy using the revised criteria was 77%. The authors concluded that the Mendoza classification increased the DSOC interobserver agreement and accuracy rate by 16% and 20%, respectively, compared to previous criteria.

Despite few studies showing the poor accuracy of visual impressions for the characterization of BSs [26,27], there is a lot of available evidence showing that visual impressions have an optimal accuracy that sometimes is superior to a targeted biopsy (the so-called “cholangioscopy paradox”). A meta-analysis conducted by de Olivera et al. of six studies showed an overall pooled sensitivity and specificity of visual interpretation of biliary malignancies by DSOC of 94% (95% CI 89–97) and 95% (95%CI 90–98), respectively [28]. An interesting recent multicenter prospective study on 289 patients with indeterminate BSs, who underwent a cholangioscopy with Digital SpyGlass, showed that strictures were visualized effectively in 98.6% of cases, in which a diagnostic visual impression was obtained in 87.2% of patients. The visual impression of malignancy showed a sensitivity of 86.7% and a specificity of 71.2% compared with the final diagnosis obtained after a 6-month follow-up [29]. Finally, a recent Italian prospective multicenter study on 369 patients evaluated the procedural success of DSOC in the setting of indeterminate BSs; the authors found a sensitivity, specificity, and accuracy for visual impressions of 88.5%, 77.3%, and 83.6%, respectively, with the gold standard as surgery or a negative follow-up after 12 or more months [30]. It should be highlighted that in peculiar clinical situations, such as patients with primary sclerosing cholangitis and/or prior plastic stent positioning, the accuracy of POC may decrease but remains superior to brush cytology [31].

Other than the possibility to macroscopically evaluate the biliary mucosa, POC has the great advantage of performing target biopsies under direct visualization. A recent prospective multicenter randomized trial showed that the sensitivity of DSOC-guided biopsies (SpyBite Biopsy) was higher than that of brush cytology during ERCP (68% vs. 21%) [32]. Also, a recent meta-analysis by Wen et al., based on 356 patients, showed the good accuracy of DSOC-guided biopsies, with a sensitivity and specificity of 74% and 98%, respectively [21]. More recently, the aforementioned study by Fugazza et al. showed a high sensitivity of up to 80% with the SpyBite [30].

To our knowledge, no data are present in the literature to date on the optimal number of biopsies to perform during POC; however, at least two biopsies are performed in most of the available studies [21].

### 2.3. Advanced Diagnostics: Artificial Intelligence and Confocal Laser Endomicroscopy

The integration of artificial intelligence (AI) in gastrointestinal endoscopy has gained popularity and has allowed its early application in clinical practice, especially in the upper and lower gastrointestinal tract for the discrimination of pre-cancerous lesions [33,34]. Confocal laser endomicroscopy (CLE) is another innovative diagnostic tool that offers real-time in vivo histopathological data during endoscopic examinations, and it is applied in several gastrointestinal fields in order to obtain a real-time diagnosis [35]. This technology employs low-energy laser light emission to generate tissue images, thereby enhancing the precision of targeted biopsies and facilitating immediate optical biopsies [36].

Application of these kinds of advanced techniques in bilio-pancreatic diseases is most useful for the discrimination of malignant and benign BSs. Despite many studies, as described above, showing the optimal accuracy of DSOC for BS determination, diagnostic evaluation with cholangioscopy remains challenging, especially for non-expert endoscopists. Moreover, a recent study showed a poor interobserver agreement between expert endoscopists in classifying images of BSs [26]. The abovementioned evidence suggests the need for more advanced techniques to integrate the diagnostic power of cholangioscopy.

Application of CLE during cholangioscopy was first described in 2008 [37]. Thereafter, several studies have been published showing a good accuracy in diagnosing malignant BSs [38]. The most recent meta-analysis that was published showed a pooled sensitivity of 0.88 and a pooled specificity of 0.79 for a final diagnosis of an indeterminate BS [39].

Despite these valid and consolidated results, application of CLE in cholangioscopy has not gained popularity and, at the moment, CLE is far from being extended to clinical practice.

In contrast to CLE, AI is increasingly gaining attention and widespread use in endoscopic applications, considering its ease of use and also valid results in randomized trials in other endoscopic fields such as colonoscopy. The application of AI in cholangioscopy is probably the most recent application of this revolutionary technology in endoscopy and some papers are starting to became available in the literature. The first pilot study on AI applied to DSOC was published in 2022 [40]. The authors developed, trained, and validated a convolutional neural network (CNN) based on DSOC images. Each frame was labeled as a normal/benign finding or as a malignant lesion if histopathologic evidence of biliary malignancy was available. The model had an overall accuracy of 94.9%, a sensitivity of 94.7%, a specificity of 92.1%, and an AUC of 0.988 in a cross-validation analysis. More recently, two other papers [41,42] have been published using the CNN system, confirming the accuracy of AI in predicting neoplastic BS; although, in a paper by Robles-Medranda et al., a lower specificity was found (68.2%) [41]. A different model of AI has been recently proposed, the “MBSDeiT”, which consists of two models to identify qualified images that are then used to predict malignant BSs in DSOC videos in real time. In this pivotal study, AI with MBSDeiT appeared promising and achieved superior performance to that of expert and novice endoscopists [43]. In Table 3, all the most relevant studies on the diagnostic role of cholangioscopy in biliary disease are summarized.

## 3. Difficult Bile Duct Stones

Choledocholithiasis represents the most common indication of ERCP [46]. Approximately 85–90% of bile duct stones can be removed according to the guidelines during a standard ERCP with balloons or basket retrieval catheters, with a comparable effectiveness and safety [47]. Conversely, the remaining 10–15% of stones are often more challenging to remove and necessitate additional or advanced techniques; these cases are referred to as difficult biliary stones (DBSs). There are different characteristics that could increase the difficulty of stone extraction, such as a size larger than 15 mm, multiple stones or square/barrel-shaped stones, and a location in the intrahepatic duct or in the cystic duct. Anatomical features of the CBDs could also contribute to a challenging stone extraction such as sigmoid-shape CBDs, narrowing of the CBD distal to the stone, acute distal CBD angulation < 135°, or a shorter length of the distal CBD [48].

In the presence of DBSs, the latest ESGE guidelines recommend limited sphincterotomy (1/3 to 1/2 of the distance to the papillary roof) combined with 30 to 60 s of endoscopic papillary large balloon dilation (EPLBD) as the first-line approach [47]. When EPLBD fails or is not indicated for the treatment of DBS, lithotripsy is the suggested approach and could be performed either mechanically, through extracorporeal shock waves, or assisted by cholangioscopy.

### 3.1. Cholangioscopy-Assisted Lithotripsy

Cholangioscopy-assisted lithotripsy can be performed using electrohydraulic or laser energy. Both techniques use a probe inserted into the operating channel of the cholangioscope. Saline solution irrigation is crucial to provide a medium for shock wave transmission, as well as to allow visualization of the duct and stones and to flush away debris [49]. Autolith Touch EHL (Nortech; Northgate Technologies Inc., Elgin, IL, USA) is a U.S. Food and Drug Administration-approved EHL (electrohydraulic lithotripsy) system that consists of a single-use probe with different power settings.

Laser lithotripsy (LL) is a laser light usually obtained with holmium with a precise wavelength that delivers an impulse to induce wave-mediated stone fragmentation. After lithotripsy, the stone fragments are subsequently extracted with standard techniques.

The efficacy of cholangioscopy-assisted lithotripsy for treatment of DBSs has been evaluated in several studies. In the meta-analysis by Korrapati and colleagues, an overall estimated stone clearance rate of 88% (95% CI 85–91%) and an estimated stone recurrence rate of 13% (95% CI 7–20%) were reported, without significant evidence of heterogeneity among the studies [50]. In a recent multicenter prospective “real-life” study, Fugazza and colleagues investigated the safety and efficacy of lithotripsy performed either with EHL or LL. They treated 152 patients for DBSs with a median size of 20 mm (range 12–45 mm) and a “difficult” localization in 23% of patients [30]. Overall, the complete duct clearance was comparable to the results of the meta-analysis by Korrapati et al. (92.1%); interestingly, in 82.1% of patients, complete bile duct clearance was obtained in one session.

In the literature, numerous trials analyze different methods for stone clearance, comparing cholangioscopy-assisted lithotripsy with other ERCP techniques. In the first randomized controlled trial (RCT) in 2018, Buxbaum and colleagues showed that cholangioscopy with LL was more effective than conventional therapies in achieving complete bile duct clearance of stones bigger than 1 cm (39/42, 93% vs. 12/18, 67%, *p* = 0.009) [51]. More recently, two other RCTs have been conducted on LL, showing a high success rate (94–100%), similar to previous studies [52,53]. It should be acknowledged that these comparative studies have certain limitations, ranging from the small sample size and variability in included patients to the use of different accessories and sub-techniques. Facciorusso A. and colleagues produced a systematic review and a network meta-analysis utilizing GRADE methodology to address the question of which method is most effective for DBS treatment. Moderate-quality evidence indicates that cholangioscopy-assisted lithotripsy was superior to the other techniques (EPLBD and mechanical lithotripsy) and ranked the highest in increasing the success rate of DBS removal (SUCRA score, 0.99) [54].

Regarding which probe (EHL or LL) performs better during lithotripsy, no prospective RCTs have been detailed. A multicenter international study involving 22 tertiary centers retrospectively included 407 patients who underwent cholangioscopy-assisted lithotripsy (306 with EHL and 101 with LL). The procedure outcomes were similar between groups. Overall, complete bile duct clearance was achieved for 96.7% in EHL and 99% in LL (*p* = 0.31) and single-session successful treatment was achieved for 74.5% in EHL and 86.1 in LL (*p* = 0.20). Notably, the mean time was longer in the EHL group (73.9 min) than in the LL group (49.9 min; *p* < 0.001) [55].

Despite the latest guidelines published in 2018 stating that there are no differences in efficacy between EHL and LL [47], more recently, some studies demonstrated results in favor of LL. In a study performed by a Dutch group, LL achieved a higher rate of bile duct clearance than EHL (405/426, 95.1% vs. 245/277, 88.4%; *p* < 0.001) and the AEs were higher in the EHL group (13.8% vs. 9.6%; *p* = 0.04) [56]. A systematic review and meta-analysis by McCarthy and colleagues showed that single-session lithotripsy success rates were higher in LL compared to EHL (83% vs. 71%; *p* = 0.021) [57]. Furthermore, the mean procedure time was significantly longer for EHL compared to LL (*p* < 0.001), despite the mean stone size of the EHL group being smaller compared to the LL group (*p* < 0.001). Similar results were reported in a meta-analysis by Amaral et al., where only prospective studies were included (DBS clearance LL vs. EHL: OR 3.09; 95% CI: 1.71–5.59) [58]. 

### 3.2. Diagnostic Role of Cholangioscopy in DBS Clearance

When DBSs are treated with EPLBD or mechanical lithotripsy, evaluation of the complete clearance of the biliary tree from biliary stones could be challenging. Usually, balloon-occluded cholangiography is performed to confirm bile duct clearance [59]. However, in patients with residual small-sized stones, a large bile duct, or pneumobilia, adequate bile duct evaluation can be difficult. Moreover, complete clearance of bile ducts is fundamental to prevent future complications [60]. For these reasons, when clearance of the bile ducts is uncertain and the CBD is sufficiently dilated, it is possible to directly evaluate the presence of residual stones with D-POC using an ultra-slim gastroscope or a standard gastroscope, according to the size of the CBD. This type of examination does not increase the global costs of the procedure and it confirms the clearance of the CBD. A retrospective study on 36 patients with DBSs treated with EPLBD during ERCP showed that D-POC after ERCP found residual CBD stones in 22.5% of patients, allowing the complete clearance of stones that were not diagnosed with balloon-occluded cholangiography [44].

## 4. Other Applications of Cholangioscopy

### 4.1. Primary Sclerosing Cholangitis

Primary sclerosing cholangitis (PSC) is a rare, chronic cholestatic liver disease characterized by intrahepatic or extrahepatic strictures or both, with bile duct fibrosis. Patients with PSC are at increased risk of developing cholangiocarcinoma. Differential diagnosis between inflammatory and neoplastic strictures has probably been the most challenging issue throughout the natural history of the disease. Surveillance strategies with MRCP, CA19-9 dosage, and ERCP sampling may have limited the diagnostic yield to detect malignancy [61].

Cholangioscopy in patients with PSC allows a direct endoscopic evaluation of the stricture and may increase the sensitivity to detecting malignancies. However, few studies are available on the topic, with small sample sizes and suboptimal evidence of efficacy. The first generation of SOC appeared promising in comparison to ERCP for detecting malignancy in patients with PSC in terms of the overall accuracy (93% vs. 55%; *p* < 0.001) [62]. More recent studies performed with DSOC showed a sensitivity between 33 and 75% in detecting cholangiocarcinoma within a dominant stricture [45,63]. Indeed, the most recent European guidelines on PSC suggest the use of cholangioscopy only in selected cases [64].

In 2019, a Canadian group proposed a classification system, the “Edmonton Classification”, for extrahepatic PSC based on the visual characteristics under direct cholangioscopic evaluation [65]. Three phenotypes were proposed based on the cholangioscopic characteristics: the inflammatory type, with mucosal erythema and active inflammatory exudate; the fibro-stenotic type, characterized by concentric fibrotic scars; and the nodular or mass-forming type, identified by a mass in the involved segment of the extrahepatic bile duct. Prospective studies for the validation of this novel classification could be of help in order to obtain a cholangioscopic pattern that correlates with malignancy.

### 4.2. Mirizzi Syndrome

Mirizzi syndrome (MS) is a rare biliary stone disease generally caused by external compression of the CBD or common hepatic duct due to an impacted gallstone within the gallbladder or cystic duct. Cholangioscopy-assisted lithotripsy has emerged as a successful treatment option since conventional ERCP often fails to remove this kind of stone.

Bhandari and colleagues reported their experience, in which 34 patients with MS and biliary stones located at the level of the cystic duct were treated with LL. Single-session ductal clearance was effective in 32 patients (94%), with a low incidence of adverse events such as fever, transient abdominal pain, and self-limited pancreatitis [66]. In another study by Tsuyuguchi et al., 50 patients with type II MS according to the McSherry classification system were treated using EHL. Complete stone clearance was accomplished in 95% of patients and during the follow-up period of 5 years, recurrence was observed in only 16% of patients with a rate of cholangitis of 6% [67].

### 4.3. Other Diagnostic and Operative Applications

Cholangioscopy has other clinical applications that are described in several case reports and case series. Internal migration of the biliary stent is a challenging situation that is usually treated during ERCP. However, fluoroscopic-guided retrieval of the stent is especially difficult when the CBD is markedly dilated and the ERCP accessories float in the duct or when the stent is migrated above the biliary confluence. Direct visualization of the stent during cholangioscopy allows its removal under endoscopic visualization [68]. Moreover, the D-POC system’s conventional endoscopic accessories (e.g., polypectomy snares and foreign body forceps) could be used for removal of the migrated stent [69].

Benign polyps or low-grade malignant lesions could grow, on rare occasions, inside the bile ducts, causing symptoms such as jaundice and/or cholangitis. In such cases, demolitive surgery is not indicated but an operative treatment is required to avoid clinical complications. Endoscopic removal of endo-biliary lesions has been described in several reports and it is a valid, minimally invasive option [70,71,72]. Malignant endoductal polypoid lesions are rarely observed, and in case of signs of biliary obstruction, debulking could be attempted during D-POC with hot snare polypectomy [73].

Radiofrequency ablation (RFA) is performed in cases of intraductal biliary lesions for therapeutic or palliative reasons. However, the effectiveness of RFA based on radiographic guidance alone may be insufficient, and when it is performed outside the target, severe complications may occur [74]. Therefore, cholangioscopy has the possibility to perform RFA under direct view [75] or to evaluate the effect of previous RFA procedures [76].

Post-ERCP bleeding is a life-threatening condition that sometimes could be difficult to manage with standard endoscopic hemostatic techniques. When endoscopic management fails, radiological embolization may be performed, but interventional radiology is not always available and embolization of branches of the pancreatoduodenal artery could also create complications. When the source of bleeding is endoductal, exploration of the CBD with the D-POC allows the identification of the source of bleeding and the execution of the most appropriate hemostasis technique [77].

Finally, biliary cannulation remains one of the most challenging steps during the ERCP procedure. Different biliary cannulation techniques and pharmacological options are used to prevent post-ERCP pancreatitis (PEP). However, among studies and in clinical practice, PEP remains one of the most feared complications [78]. A recent pilot study by Liu et al. showed the feasibility of cholangioscopy-guided biliary cannulation under direct endoscopic visualization of the papilla. Selective cannulation of the CBD under the endoscopic visualization of DSOC was feasible and safe [79]. This kind of cannulation technique could be promising, especially in difficult anatomic situations (e.g., intradiverticular papilla) or when the use of X-rays could limit the use of ERCP (e.g., pregnancy).

## 5. Cholangioscopy through Different Routes

Cholangioscopy is traditionally performed through the standard retrograde transpapillary access during ERCP or with the D-POC. Nevertheless, in some situations, the transpapillary route is not accessible [80]. One of the most common reasons for unreachable papilla is an altered upper gastrointestinal anatomy related to previous surgical intervention. Roux-en-Y reconstruction is the most unfavorable anatomy because the afferent limb is usually too long to reach with the duodenoscope and consequently with the cholangioscope [81]. It should also be acknowledged that in 2011 the global total number of bariatric surgeries was approximately 340.000, and among them, Roux-en-Y-Gastric Bypasses (RYGB) were by far the most commonly performed procedure. About one-third of post-bariatric patients develop gallstones; therefore, the number of patients with RYGB that require biliary drainage is consistent [81,82].

Biliary cannulation can also be hampered by anatomical conditions such as an intradiverticular papilla, which may increase the rate of failed cannulation [83]. Transpapillary access could also have some limitations for some therapeutic applications of POC, for example, when intrahepatic stones are located in the more proximal branches of the biliary tree [84].

For these reasons, alternative cholangioscopic access points are possible and could be considered case-by-case according to the clinical indication. Below, we present all the possible alternative cholangioscopic access points in the relative literature.

### 5.1. Percutaneous

Percutaneous cholangioscopy (PC) was firstly described in 1983, when the first case series of 39 patients was published by Gazzaniga et al., showing that the procedure was safe and effective for removing biliary stones [85].

The PC procedure requires the preliminary execution of percutaneous transhepatic biliary access with the standard interventional radiology technique. Once the percutaneous biliary tube is in place, a preliminary cholangiogram is measured to confirm its location and a guidewire is advanced through the biliary tube and into the small bowel. Then, the biliary tube is removed and a tract dilation is usually performed in order to permit access to a 12F or 16F sheath. Cholangioscopy is therefore performed through the percutaneous catheter [86].

PC could be performed with the same endoscopes used for POC [32]; however, the standard cholangioscopes used for POC are too long to pass through the short transhepatic route, and they could therefore be cumbersome to manage. Recently, a novel disposable short digital cholangioscope (SpyGlass™ Discover, Boston Scientific Endoscopy, Marlboro, MA, USA) of 65 cm in length has become available on the market. This short cholangioscope is easier to manage and several case reports have shown its efficacy in treating difficult bile stones and retrieving migrated stents from the percutaneous route [87,88]. Another short cholangioscope is available (CHF-CB30 short, Olympus, Tokyo, Japan) and its feasibility has been demonstrated for percutaneous use [86]; however, it is a fiberoptic scope with limited maneuverability with only two directions of movement.

The indications for PC are both diagnostic and therapeutic. The main diagnostic indication is the assessment of indeterminate BSs. The therapeutic indications include difficult bile stones, unreachable papilla (e.g., in the case of altered anatomy), and failed papilla cannulation. The data in the literature are limited to several case reports and small case series. The largest multicenter series on 28 patients was published 2021; the majority of patients had an altered post-surgical anatomy (25/28 patients) and PC was technically successful in one session in 96% of patients [89]. The majority of patients successfully received lithotripsy for biliary stones and five malignant strictures were found with a histology accuracy of 100% and a visual impression sensitivity of 83.3%. Another small case series of four patients demonstrated the efficacy of PC in visual and histological diagnoses of indeterminate BSs [90].

Post-orthotopic liver transplantation (OLTx) BSs are a common complication that may occur in both preserved anatomies (stricture at the level of biliary anastomosis of the recipient and donor) and altered anatomies (stricture at the level of hepatic-jejunal anastomosis) and may require multiple endoscopic interventions (e.g., endoscopic multistenting) [91]. However, endoscopic multiple access in the case of an altered anatomy could be time consuming and challenging [92]. A recent retrospective study demonstrated the efficacy of PC for the treatment of post-OLTx BSs, with failure in only 2 out of 25 patients [93]. Thus, PC could be considered as an alternative to device-assisted ERCP for this indication.

POC could also be challenging in the case of normal anatomies and successful biliary cannulation when biliary stones are located in the more proximal biliary branches, where biliary angulations may block POC passage or may create instability during therapeutic procedures. In such situations, PC may provide valid and easier access to intrahepatic/hilar bile ducts for the treatment of biliary stones [94,95,96,97,98]. In Figure 2, a case from our institution of an EHL for multiple intrahepatic lithiasis not reachable with POC for the presence of an inflammatory stricture below the stones is described.

### 5.2. Trans-Cystic

Intraoperative exploration of CBD is an old procedure that is performed by surgeons during cholecystectomy for the clearance of concomitant choledocholithiasis. Surgical CBD stone clearance has demonstrated its efficacy in several previous trials [99], but the use of older choledochoscopes is difficult and not routinely performed by all surgeons [100]. In addition, the rapid development and wide diffusion of ERCP has led to a further reduction in the use of surgical choledochoscopes [101].

In the case of concomitant gallbladder and CBD lithiasis, ERCP is usually performed before cholecystectomy or during surgery with the rendez-vous technique [102]. However, ERCP with the rendez-vous technique is uncommon in routine clinical practice because of organizational drawbacks such as the availability of the endoscopic and surgical equipment. On the other hand, performing ERCP before cholecystectomy may prolong the hospitalization stay [103] and in some clinical situations, such as acute cholecystectomy, the timing of surgery is critical and the presence of a concomitant choledocholithiasis may delay cholecystectomy [104].

The most recent cholangioscopes have the possibility to explore and clear the CBD through a transcystic approach during laparoscopic cholecystectomies. Some case reports have been published with the first generation of cholangioscopes, showing efficacy in clearing CBDs with the use of lithotripsy [105,106].

The advent of short digital cholangioscopes (SpyGlass™ Discover, Boston Scientific Endoscopy, Marlboro, MA, USA) may improve the endoscopic vision inside the bile duct and also their maneuverability by the physician. The first report of intraoperative exploration of CBDs with a short cholangioscope was published in 2020 by Palermo et al. [107]. Recently, preliminary data on the use of short SOCs have been published by our group; ten patients with acute cholecystitis and concomitant choledocholithiasis were treated with a trans-cystic short SOC, showing a technical success of 100% of cystic duct cannulation and ability to clear the stones in all of the eight patients with CBD stones. The procedure was also safe, with only one case of mild acute pancreatitis [108]. It should be noted that a short SOC was used by surgeons without the need for endoscopists in nine out of the ten cases. Therefore, a trans-cystic SOC could be applied in specific situations when surgical timing is critical (e.g., acute cholecystitis), when papilla are not reachable in an altered surgical anatomy, or when routine organization may limit the performance of ERCP. In Figure 3, a laparoscopic trans-cystic cholangioscopy is shown.

### 5.3. Trans-Hepaticogastrostomy

Therapeutic biliopancreatic endoscopy has had an impressive evolution in the last decade, especially with the expansion of interventional EUS. Biliary drainage can now be performed with different EUS-guided techniques such as choledochoduodenostomy, hepaticogastrostomy (HGS), and gallbladder drainage [109]. HGS is a challenging technique that allows for transgastric or transjejunal biliary drainage, according to the specific anatomy of the left biliary segments [110]. This type of drainage is more physiological than percutaneous drainage and has less long-term adverse events [111,112]. HGS is usually performed in the case of malignant obstruction of both hilar or distal strictures when standard endoscopic drainage techniques are not feasible (duodenal obstruction, failed ERCP, and altered anatomy) [110]. HGS is actually performed with dedicated metal stents that have a diameter of eight or ten mm [113]. This type of dimension allows passage of different devices and also cholangioscopes through the stent.

In the case of suspected malignant obstruction, HGS is associated with less adverse events than percutaneous biliary drainage, and therefore diagnostics investigations could be performed through the HGS stent [114]. Indeed, some case reports have shown the ability of POC for diagnosing malignancies with direct biopsies or with a visual impression. The procedure is safe, without increased risk of stent displacement [115,116,117].

In the case of benign indications, HGS compared to percutaneous access has the possibility to perform multiple endoscopic revisions as the stent reduces the risk of infection of the percutaneous access with a better long-term tolerability. Patients with hepaticojejunal anastomosis strictures or huge intrahepatic stones may require multiple endoscopic interventions. The application of POC through the HGS route has been described in various case reports and it was effective in performing lithotripsy and stone removal [118,119,120,121,122,123,124,125,126] and for the treatment of anastomotic strictures [127,128,129]. POC through HGS also allowed for the removal of migrated biliary stents in two case reports [130,131].

## 6. Conclusions

Cholangioscopy is a technique that has been present in clinical practice for several decades, but has only entered routine practice after device optimization toward single-operator use and improvements in endoscopic images. Compared with other biliopancreatic endoscopic techniques, cholangioscopy has the great advantage of directly visualizing the biliary lumen and mucosa. Therefore, cholangioscopy is complementary to ERCP and EUS for the evaluation of challenging clinical situations such as indeterminate biliary strictures or complex choledocholithiasis. In such cases, a multidisciplinary approach with surgeons, oncologists, and radiologists is fundamental to evaluate the optimal diagnostic and therapeutic approach.

A strength of cholangioscopy is its reproducibility in the evaluation of biliary mucosal patterns [25]. This aspect should not be underestimated because it is a prerequisite for the widespread use of this advanced technology, allowing endoscopists to “speak the same language” when they face a biliary stricture. In addition to reproducibility, cholangioscopy is highly effective after the failure of other advanced techniques (e.g., tissue acquisition during ERCP/EUS-FNB or standard stone extraction), avoiding the need for more invasive procedures such as surgery or incorrect treatment allocation (e.g., surgery for a benign biliary stricture).

Cholangioscopy can be performed through different access points: transpapillary, percutaneous, trans-cystic, or through a stent. This expands the field of application (e.g., to patients with altered anatomies), with the aim to improve patients’ outcomes and to reduce the need for invasive procedures. Finally, cholangioscopy has demonstrated its safety in both diagnostic and therapeutic applications.

One of the main critiques that usually is made of cholangioscopy is related to the cost of the device. However, it should be noted that cholangioscopy is usually applied to complex patients that have already been subjected to other biliopancreatic endoscopic procedures. Therefore, early application of cholangioscopy could reduce the global procedural cost. An economic study by Deprez et al. showed that the use of SOC for both complex choledocholithiasis and BS determined a decrease in the number of procedures (−27% and −31% relative reduction, respectively) and costs (−EUR 73,000 and −EUR 13,000, respectively) when compared with ERCP [132]. Moreover, a cost-effectiveness model was developed, aiming to determine the timing of SOC-EHL introduction in the management of choledocholithiasis. In this study, early utilization of EHL for DBSs was the cheapest strategy, with an effectiveness similar to one or more conventional ERCP procedures with delayed lithotripsy [133].

Difficulty in cholangioscopic execution is another interesting factor but the most recent devices have improved maneuverability. The use of SOC has also eliminated the concomitant need for two operators.

In conclusion, the availability of cholangioscopy is recommended in endoscopic centers in order to investigate and treat the most challenging clinical situations with the aim of reducing the diagnostic delay for indeterminate BS and treating situations such as complex choledocholithiasis in challenging scenarios such as altered anatomies.

## Figures and Tables

**Figure 1 diagnostics-13-02933-f001:**
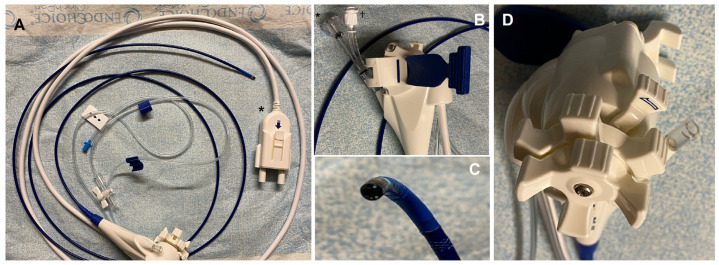
Digital version of the single-operator cholangioscope (DSOC) (SpyGlass, Boston Scientific Endoscopy, Marlboro, MA, USA). (**A**) The cholangioscope in its full length (214 cm) attached with a catheter cable (white catheter) to the cable connector (black star). (**B**) Details of the attachment strap (blue part), which allows for the cholangioscope to be fixed to the shaft of the duodenoscope; the Y-port adapter, with a working channel for accessory access (black star); and the irrigation–aspiration port (black cross). (**C**) Details of the tip of the cholangioscope, featuring two LED lights and two irrigation channels. (**D**) Details of the two wheels capable of four types of movement and the articulation lock.

**Figure 2 diagnostics-13-02933-f002:**
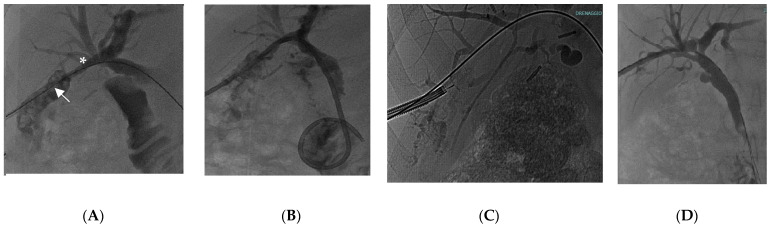
56-year-old female with intrahepatic multiple lithiasis of the sixth segment. Per-oral cholangioscopy was not feasible for the presence of an inflammatory stricture below the stones. Percutaneous cholangioscopy was performed with complete clearance of stones. (**A**) Multiple stones in the sixth liver segment (white arrow) above the biliary stricture (white asterisk); (**B**) percutaneous drainage; (**C**) radiological view of percutaneous cholangioscopy with electrohydraulic lithotripsy probe; (**D**) percutaneous cholangiography showing complete clearance of the sixth biliary segment.

**Figure 3 diagnostics-13-02933-f003:**
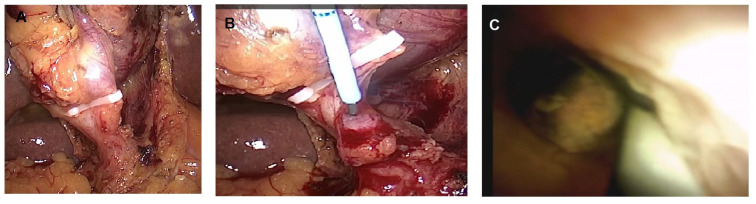
Laparoscopic trans-cystic cholangioscopy with a small stone in the distal common bile duct. (**A**) Clipped cystic duct; (**B**) cholangioscopy insertion in a guidewire inside the cystic duct; (**C**) cholangioscopic view of the small stone that was removed with a retrieval basket.

**Table 1 diagnostics-13-02933-t001:** Technical characteristics of the available cholangioscopy systems.

	Mother–Baby Scope System	Direct Videocholangioscopy with Ultraslim Gastroscope (D-POC)	Digital Single-Operator Cholangioscopy (D-SOC)		Short D-SOC	
	EVIS LUCERA ELITE Video Cholangioscope Olympus		SpyGlass™ DS II Direct Visualization System Boston Scientific	eyeMAX Micro-Tech™	SpyGlass™ Discover Boston Scientific	Olympus CHF-CB30S
**Required endoscopist**	2 operators	1 operator	1 operator	1 operator	1 operator	1 operator
**Outer diameter**	3.4 mm	5–5.9 mm	3.6 mm	3–3.6 mm	3.6 mm	2.8 mm
**Accessory channel**	1.2 mm	2 mm	1.2 mm	1.2–2 mm	1.2 mm	None/
**Length**	192 cm	110–170 cm	286 cm	220 cm	65 cm	70 cm
**Angulation function**	Two way, 70°	Four way, up to 210°	Four way, 30°	Four way, 30°	Four way, 45°	Two way, 120°
**Quality of images**	Enhanced Near-Point Image Quality	High-definition resolution	High-definition resolution	High-definition resolution	Full HD	Fiber-optic
**Image-enhanced function system**	Available	Available	Not available	Not available	Not available	Not available

D-POC, direct peroral cholangioscopy.

**Table 2 diagnostics-13-02933-t002:** Etiologies of benign and malignant biliary strictures.

Malignant Aetiologies	Benign Aetiologies
Pancreatic adenocarcinoma	Post-surgical iatrogenic stenoses (post-cholecystectomy or post-liver transplantation)
Cholangiocarcinoma (sporadic or PSC-associated)	Inflammatory forms (PSC, IgG4-associated cholangitis, sarcoidosis, histiocytosis X, eosinophilic cholangitis)
Metastatic cancer	Chronic pancreatitis
Lymphoma	Infectious diseases
Hepatocellular cancer	Vascular diseases (ischemic cholangiopathy, vasculitis, etc.)
Gallbladder cancer	AIDS cholangiopathy
	Cholelithiasis (Mirizzi syndrome)

PSC, primary sclerosing cholangitis.

**Table 3 diagnostics-13-02933-t003:** Most relevant studies on the diagnostic role of cholangioscopy in biliary disease.

First Author	Year of Pubblication	Number of Patients	Typology of Study	Main Results of the Study
Sethi A. [23]	2022	40	Two-phase validation study	Validation of the Monaco Classification for the use of DSOC in BS. Eight visual criteria were identified and global diagnostic accuracy was 70%.
Robles-Medranda C. [24]	2018	171	Observational, analytical, case-crossover, ambispective, diagnostic study	Validation of a novel classification for the use of DSOC in BS. High reproducibility (k > 80%) and sensitivity of 96.3% for neoplastic diagnosis.
Kahaleh M [25]	2022	50	Validation study	Validation of the Mendoza classification for the use of DSOC in BS. Five revised visual criteria were identified. Agreement was almost perfect and the overall accuracy was 77%.
De Oliveira P [28]	2020	283	Systematic review and meta-analysis	The overall pooled sensitivity and specificity of DSOC in the visual interpretation of malignant BS were 94% and 95%.
Fugazza A [30]	2022	381	Prospective, multicenter	This real-world study evaluated the efficacy of POC for evaluation of BS and treatment of DBS. Overall procedure success was 96.7%. Sensitivity of visual impression for malignant BS was superior to cholangioscopy-guided biopsies (88.5% vs. 80.2%).
De Vries A.B [31]	2020	80	Retrospective, single center	This study highlighted the limited diagnostic use of POC in specific situations such as PSC and previous placement of biliary stent.
Gerges C [32]	2020	60	Randomized controlled trial	DSOC-guided biopsy samples were superior to ERCP-guided brushing (68.2% vs. 21.4%) for diagnosis of malignant hilar stricture.
Saraiva MM [40]	2022	85	Pilot validation study	The authors developed, trained, and validated a CNN based on DSOC images. The model had an overall accuracy of 94.9% for differentiating malignant BS from benign BS.
Robles-Medranda C [41]	2023	Phase 1: 48 patients Phase 2: 116 patients	Two-stage validation study	Validation of a CNN model for identification of neoplasia in indeterminate BS. The model showed a good accuracy in distinguishing neoplastic lesions.
Zhang X [43]	2023	150	Multicenter diagnostic study	Validation of a novel AI model, MBSDeiT, which accurately identified 92.3% of malignant BS in prospective testing videos.
Anderloni A [44]	2019	36	Retrospective, single center	This study showed the efficacy of D-POC for the evaluation of complete clearance of CBD after removal of DBS.
Arnelo U [45]	2015	47	Prospective, single center	Study performed on patients with PSC showing the limited sensitivity of DSOC for malignant BS.

DSOC, digital single-operator cholangioscopy; BS, biliary stricture; POC, peroral cholangioscopy; CNN, convolutional neural network; CBD, common bile duct; DBSs, difficult bile stones; PSC, primary sclerosing cholangitis.

## Data Availability

Not applicable.

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
