# Peer review of "The Role of Cholangioscopy in Biliary Diseases"

_diagnostics, 2023, doi:10.3390/diagnostics13182933_

Round 1

Reviewer 1 Report

In this manuscript, the authors review the available evidence concerning the use of cholangioscopy in the management of biliary diseases. 

In particular, the authors investigate the current role of cholangioscopy in the management of difficult biliary stones untreatable with ERCP, indeterminate biliary strictures, and in particular situations like mirizzi syndrome, primary sclerosing cholangitis, intraluminal biliary benign tumors.

In addition, the authors review available literature regarding the comparison between ERCP and cholangioscopy for the management of above mentioned pathological situations.   

the manuscript is overall well written, well organized, interesting. 

I only have some minor comments: 

I suggest the authors to carefully check the manuscript: I could find some grammar mistakes, like (line 201) diagnostical should be replaced by diagnostic, (line 206) diagnose should be replaced by diagnosing, (line 250) the comma should be deleted, (line 207) delete "is", (line 329) allowS, (341) replace distinguished with characterized. 

Section 5. Cholangioscopy through different access: I would mention here, as an increasingly encountered situation that may be responsible for unreachable papilla due to upper gastrointestinal altered anatomy related to previous surgical intervention, the bariatric surgery. 

Good. 

Author Response

We thanks the reviewer for his/her constructive comment.

We modified the grammar mistakes.

We agree with the reviewer that biliary drainage in post-bariatric surgical anatomy patients is an hot-topic and we added a comma in section 5 (line 409-411).

Reviewer 2 Report

Excellent review on a cutting-edge topic. Maybe a table summarizing the latest study on the diagnostic use of cholangioscopy could be usefult to the reader.

Reference 19 is not pertinent. I suggest to replace with another relevant study assessing newer FNB needles (PMID: 35124072)

Author Response

We thank the reviewer for his kind and constructive comments.

We added the requested table on the most recent and relevant studies on the diagnostic use of cholangioscopy. We then modified the reference number 19.

Reviewer 3 Report

This is a very meaningful review. The author aim to explore the most recent advances of the diagnostic and therapeutic role of cholangioscopy for the management of biliary diseases.

What are the potential applications of AI in the direction of choledochoscopy in the future?

At present, the most common complication of ERCP is post-ERCP pancreatitis. what will choledochoscopy change for PEP prevention?

Author Response

This is a very meaningful review. The author aim to explore the most recent advances of the diagnostic and therapeutic role of cholangioscopy for the management of biliary diseases.

What are the potential applications of AI in the direction of choledochoscopy in the future?

We thank the reviewer for his/her constructive comments.

Considering the application of AI during cholangioscopy we showed in the paper that in the most recent year several pilot studies (ref. 40-43) demonstrated the efficacy of AI in the identification of malignancy in the context of indeterminate biliary stricture.

At present, the most common complication of ERCP is post-ERCP pancreatitis. what will choledochoscopy change for PEP prevention?

We thank the reviewer for this very interesting comment. Very recently a paper have been published (Liu WH, Endoscopy. 2023 Jun 20. doi: 10.1055/a-2113-8952) about the role of cholangioscopy for the incannulation of papilla under direct endoscopic vision. In this pilot study the author showed that incannulation under endoscopic vision of cholangioscopy was feasible and safe and therefore it could be a useful tool in case of difficult papilla (e.g. intradiverticular papilla) in order to reduce the incidence of PEP. A short paragraph has been added in section 4.3.

Round 2

Reviewer 1 Report

I think the authors adequately answered to my comments.  

Reviewer 2 Report

The revised version of the paper is OK. Thank you!

Reviewer 3 Report

修订版回答了我的问题,我同意发表。